# Adaptation and validation of the Physical Activity Questionnaire for Older Children (PAQ-C) among Czech children

Roman Cuberek[1]*, Marcela Janíková[2,3], Jan Dygrýn[1]

**1** Institute of Active Lifestyle, Faculty of Physical Culture, Palacký University Olomouc, Olomouc, Czech Republic, **2** Department of Sport Pedagogy, Faculty of Sports Studies, Masaryk University, Brno, Czech Republic, **3** Research Incubator in Sport Sciences, Faculty of Sports Studies, Masaryk University, Brno, Czech Republic

☯ These authors contributed equally to this work.

* roman.cuberek@upol.cz

**Data Availability Statement:** Data are available from the Mendeley Data database (http://dx.doi.org/10.17632/nmcc7wrysd.2).

**Funding:** This study was supported by the following grants—The First Stage of PAQ-C and

## Abstract

The study aimed to adapt the standardized Physical Activity Questionnaire for Older Children (PAQ-C) into the Czech language and assess its psychometric properties. A forwards-and-backwards translation method was carried out to prepare the Czech PAQ-C version (PAQ-C/CZ), followed by adjustments based on socio-cultural aspects. In the next phase, participants (n = 169) wore accelerometers for seven days. On the 8th day, participants completed the PAQ-C/CZ in school so that it was possible to determine the concurrent validity (correlation between the accelerometer and PAQ-C/CZ data, Spearman's r), internal consistency (Cronbach's alpha), item/scale properties (corrected item-total correlations, CITCs), and factor structure (exploratory factor analysis, EFA) for PAQ-C/CZ. In the last phase, participants (n = 63) completed the PAQ-C/CZ twice on two consecutive school days in the morning to determine the test-retest reliability (intraclass correlation coefficient, ICC; independent sample T-test). The PAQ-C/CZ indicated moderate internal consistency (alpha = 0.77), acceptable item/scale properties (CITCs = 0.29–0.61), and good test-retest reliability (ICC = 0.73–0.94). The EFA findings suggested a single factor model (factor load = 0.32–0.73) for PAQ-C/CZ, with items 2, 3, and 4 contributing low to the total score. Results on concurrent validity showed low but significant association (r = 0.28, p < 0.05) between accelerometer and PAQ-C/CZ data. Considering the study results, the PAQ-C/CZ can be recommended as a tool for moderate-to-vigorous physical activity assessment in large-sample research studies only, but with an emphasis on the interpretation of the correct results.

## Introduction

Due to an ample amount of research into physical activity (PA) in school-aged children and youth in recent years, the benefits of PA are well-documented. The key finding is that movement behaviour at this age influences the behaviour in adulthood considerably [1–3]. What transpires is that PA brings numerous benefits to school-aged children and youth, such as

PAQ-A Validation (No. MUNI/51/03/2019) and the Physical Activity, Cognitive Functions, and Physical Fitness in Children Aged 10–11 Years (No. ROZV/24/01/FSpS/2018) provided by the Masaryk University, Brno, Czechia. The funders had no role in study design, data collection and analysis, decision to publish, or preparation of the manuscript.

**Competing interests:** The authors have declared that no competing interests exist.

improved cardiorespiratory fitness, muscular strength, and mental health, and lowers the risk of overweight, obesity, and metabolic syndrome related to other diseases [4]. It is considered a critical supportive factor in developing cognitive function, positively impacting academic achievement besides [5]. The aforementioned benefits of PA are crucial for creating and evaluating the government's intervention programmes aimed at educational and health areas. This requires regular monitoring of the current state of PA in school-aged children and youth.

*Measuring PA is a challenging task as movement is a complex, multi-dimensional human behaviour* [6], warranting a multi-data approach. While many objective PA assessment techniques exist, they tend to be unsuitable for large-sample surveys [7]. That is why, in such cases, self-reporting questionnaires are chosen instead as they are inexpensive, easy to administer, and suitable to cover numerous aspects of PA. Unfortunately, their use in children is often limited compared to adults. The younger the child is, the higher the risk of misinterpretation or misunderstanding of the question. Moreover, children are more prone to inaccurate recall of their own past behaviour.

In the last 30 years, national data on PA and physical fitness of Czech children have been hardly comprehensive [8, 9]. Due to the limited evidence on current PA, both Czech and European institutions have no basis for policies in the field of movement behaviour in Czech children. In 2021, the Czech Ministry of Education, Youth and Sports intends to launch a national survey on PA and physical fitness in Czech children. Two questionnaires in Czech-language with modifications are currently available for this purpose of PA assessment (the Health Behaviour in School-aged Children Questionnaire and the Youth Activity Profile one). That said, neither questionnaire is fully suitable for numerous reasons. Those include time demands, the nature of their outputs, the requirements for reading and text comprehension, their psychometric qualities, etc. [10–13]. An expert panel ranked the Physical Activity Questionnaire for Older Children (PAQ-C) as one of the very few self-reported instruments that has acceptable validity, reliability, and practicality to be used in children [14]. To date, more than ten language adaptations or modifications of the PAQ-C exist [15–25]. Some of them were developed for conducting the European ALPHA study and, as such, guarantee a high standard for cross-European comparisons. For all these reasons, the PAQ-C should be considered an appropriate PA assessment tool for the Czech Ministry of Education, Youth and Sports' purposes mentioned above. This study thus aimed to adapt the PAQ-C questionnaire into the Czech language and to determine the psychometric properties of the Czech PAQ-C version (PAQ-C/CZ) in Czech children aged 10–13 years. In that sense, we examined the internal consistency, the item/scale relationship, the test-retest reliability, factor structure, and concurrent validity.

## Methods

Initially, the PAQ-C was adapted to the Czech language. After that, the psychometric properties of the PAQ-C/CZ were determined in two phases.

### Adaptation of the PAQ-C

The PAQ-C questionnaire was a ten-item self-administered 7-day recall questionnaire for children aged 8–14. The last item was not included in the total score (it identified students whose activities were unusual during the week preceding the survey). PAQ-C/CZ total score was a mean of the score of 9 items and ranged from 1 to 5. An increasing value reflected a higher level of PA. The questionnaire was intended to measure PA with moderate to vigorous intensity (MVPA). In comparison with other self-assessment tools for measuring PA, the PAQ-C proved to correlate significantly with other instruments [26].

In accordance with the guidelines for translating and validating a questionnaire [27, 28], the PAQ-C was translated to Czech by two independent professional translators and then translated back by another independent professional translator who was not familiar with the original instrument. The final versions were compared. In cases of discrepancies, a consensus was found based on a discussion. Subsequently, a 10-member expert committee comprising psychologists, sports pedagogues, sociologists, and primary school teachers was established to assess the content relevance, taking into consideration the Czech cultural specifics with respect to the intention of the original version [29]. The PAQ-C/CZ differs from the original version in several aspects. In particular, the original *spare-time activity* checklist was modified, the *recesses* and *lunch* items were merged, and a new *before-school activity* item was added to better represent the Czech culture. All modifications in the activity checklist were done in accordance with other studies on the most frequented PA or on other PA specifics in Czech children [30, 31]. For this reason, *rowing/canoeing* was removed, six activities (*combat sports*, *gymnastics*, *handball/dodgeball*, *horse riding*, *parkour/street workout*, and *fitness/yoga*) were added, and several activities were clustered (based on movement similarities in activities). Additionally, primary school teachers assessed the semantic aspects, and the modified version of the questionnaire was evaluated in two focus groups of children aged 10–12 years. These children were encouraged to discuss all unintelligible or misleading statements, questions, or any other difficulties they may have faced. All the findings were taken into account in the last revisions as the final version of PAQ-C/CZ was developed (see the S1 and S2 Files). The PAQ-C/CZ total score was calculated in accordance with the original scoring. However, the raw scores for the first item (*spare-time activity*; Q1) were rescaled using a formula from Janz et al. [32] to reflect the ranges of the other questions. We used a formula $Q1(i)_{corrected} = 4*(Q1(i)_{raw} - 1) / [Max(Q1_{raw}) - 1] + 1$, where $Q1(i)_{raw}$ was an individual Q1 raw score and $Max(Q1_{raw})$ was a Q1 maximal raw score in a sample.

## Validation of the PAQ-C/CZ

**Participants and procedures.** The Czech Republic is a rather small country with a population of approx. 10 million, a unified educational system, and a social-economic status comparable across the population. The Czech Republic is a relatively small country with a population of approx. 10 million, a unified educational system, and a social-economic status comparable across the population. In selecting the sample of schools, the municipality's size (large, medium, or small) was taken into account as this is a factor that corresponds with the availability and opportunity of partaking in various types of PA. It also reflects the diverse social-economic environments. Regarding the study's aim, we consider such a sample adequately representative of the country's school-children and youth population. The ratio of 20 subjects per item was employed to determine the number of participants to be included. Taking into consideration the nine items comprising PAQ-C and the estimated 20% drop-out, a minimum of 216 participants was determined as a suitable study population. The sizes of municipalities were considered in the selection of schools. In contrast to the original version of PAQ-C, children aged 10–13 years representing both sexes were contacted to participate in the survey. Based on the recommendations of both the expert board and their teachers, the lower age limit was moved from eight to ten years—in consideration of the children's cognitive maturity, the text's content, the level of reading comprehension, and the ability to concentrate in children of this age. The maximum age limit was set to the age of 13. We estimated that if the questionnaire was comprehensible to this age group, it could be assumed that older pupils would find it understandable as well, based on their cognitive maturity and the subsequent increase in the

level of the aspects mentioned above. PAQ-C/CZ was validated for children of the 10+ age group.

In total, 237 subjects were recruited. Participation in the study was voluntary. All participants and their parents were informed about the study details. Written and informed consents signed by parents or guardians were obtained from all the children participating in the study. The Research Ethics Committee of Masaryk University (No. EKV-2018-042) approved the study.

Initially, the internal consistency, item/scale relationship, factor structure, and concurrent validity of the PAQ-C/CZ were assessed. In the following phase, the test-retest reliability was evaluated.

*Phase 1*. During an informative meeting at school, participants were meticulously instructed on how to use the accelerometer. The next day, children started wearing the accelerometer monitors for 7 consecutive days. On the 8th day since the informative meeting, children returned the accelerometers and completed the PAQ-C/CZ under the supervision of teachers.

*Phase 2*. On two consecutive school days, children completed the PAQ-C/CZ twice, each time during the morning lesson at school. They were asked to report on the same days (i.e. on the same activities).

**Measure.** Sociodemographic data (age and sex) and anthropometric data (height in cm, weight in kg) were self-reported by participants. Anthropometric data were used to calculate participants' body mass indices (BMI, $kg/m^2$).

The ActiGraph accelerometer, model wGT3X-BT (ActiGraph LLC., Pensacola, FL, USA), was selected to objectively measure the habitual PA level. This device is a widely accepted tool for the assessment of PA in research studies [33, 34]. The accelerometer was worn on the waist (on the right side) with the exception of time spent in water or during sleep. The software Acti-Life 6.13.4 was used to initialize the devices, download, and process accelerometer data [35]. Raw data were aggregated to 15-second epochs with a normal filter. Non-wear time was defined as 60 minutes constituted of continuous 0 counts per minute, allowing for 2 consecutive minutes of counts per minute showing values of 0–100 [36]. Evenson's cut-off point values (i.e., MVPA >2295 counts per minute), which are considered the most accurate for classifying PA intensity in this age group [37], were used to assess activity intensity. A valid day was defined as valid wear-time of over 10 hours. On average, children wore the device for 13.1 ± 1.1 hours per day. A minimum of 4 valid days (3 weekdays and 1 weekend day) were needed to include the data in the final analysis [38].

**Statistical analysis.** Descriptive analyses of items were conducted for the total group and then separately for boys and girls. Overall gender effect was investigated using two-way multivariate analyses of covariance (MANOVA). Gender differences for scores in all nine PAQ-C items and for the PAQ-C/CZ total score were investigated separately using independent sample T-test. Cronbach's α was used to assess the internal consistency of the PAQ-C/CZ for the reliability analysis. Values >0.70 were considered as acceptable for general research purposes [39]. The item/scale relationship was examined by corrected item-total correlations (CITCs). CITC values >0.20 indicated a homogeneous scale [40]. Exploratory factor analysis (EFA) employing a maximal likelihood extraction method with a direct oblimin rotation was performed to identify a factor structure of the PAQ-C/CZ and to provide evidence for construct validity. Kaiser's criterion was used to determine the number of factors to be retained (eigenvalue greater than one) [41]. Construct validity in EFA can be established if items cluster into meaningful groups that reflect the behavioural domain or the theoretical constructs the questionnaire was designed to measure. In accordance with the guideline of reliability assessment [42], we determined the intra-class correlation coefficient (ICC) and its 95% confidence interval (95% CI) applying the two-way mixed effect model and the absolute-agreement type to

assess a test-retest reliability of the questionnaire. Spearman's rank correlation coefficient and its 95% CI were used to assess the concurrent validity of the PAQ-C/CZ with accelerometer data (percentage of time spent in MVPA). ICC and Spearman's values <0.50 were interpreted as poor, 0.50–0.75 were interpreted as moderate reliability, 0.75–0.9 were interpreted as good reliability, and >0.9 were interpreted as excellent reliability [43]. Student's paired T-test was used to evaluate the test-retest differences in self-reported PA. Data were analysed using SPSS 22 (IBM corp., Armonk, NY, USA). In all analyses, a significance level was set at $p < 0.05$.

## Results

### Descriptive statistics for PAQ-C/CZ and accelerometer

The final sample consisted of 206 children (aged 10–13 years; 100 girls) participated in the study; in addition to that, 31 subjects were excluded due to incomplete data. A total of 169 (age $11.0 \pm 0.7$ years; BMI $17.6 \pm 2.8$ kg/m$^2$; 81 girls) and 63 (age $11.6 \pm 0.9$ years; BMI $18.5 \pm 3.1$ kg/m$^2$; 34 girls) children were included in phases one and two of the study, respectively. 26 children participated in both phases. In the assessment of concurrent validity, 52 children were not included in the analysis for invalid accelerometer data (see Instruments).

All descriptive statistics for age, BMI, PAQ-C/CZ, and accelerometer variables are presented in Table 1. In PAQ-C/CZ score, 2 × 9 MANOVA (gender, items) did not indicate

**Table 1. Descriptive statistics for the investigated variables in the study (mean ± SD), gender differences (independent sample T-test), and test-retest differences (Student's paired T-test).**

| Variable | Phase one (n = 169; 81 girls) | | | | | Phase two (n = 63; 34 girls) | | | |
|---|---|---|---|---|---|---|---|---|---|
| | Descriptive statistics | | | Gender differences | | Descriptive statistics | | Test-retest differences | |
| | Overall | Girls | Boys | t | P-value | Test | Retest | t | P-value |
| **Age (year)** | 11.0±0.7 | 11.0±0.7 | 10.9±0.8 | 0.808 | 0.421 | 11.6±0.9 | – | | – |
| **Body mass index (kg/m$^2$)** | 17.6±2.8 | 17.0±2.6 | 18.2±2.9 | −2.982 | 0.003* | 21.8±15.1 | – | | – |
| **PAQ-C/CZ** | | | | | | | | | |
| PAQ-C total score | 2.7±0.6 | 2.7±0.5 | 2.7±0.6 | −0.482 | 0.631 | 2.6±0.6 | 2.6±0.6 | 1.212 | 0.230 |
| PAQ-C total score[a] | 2.8±0.6 | 2.8±0.5 | 2.8±0.7 | −0.518 | 0.605 | 2.7±0.6 | 2.7±0.7 | 0.806 | 0.423 |
| Spare-time activity | 1.3±0.2 | 1.3±0.2 | 1.4±0.2 | −0.608 | 0.544 | 1.3±0.2 | 1.3±0.2 | 2.537 | 0.014* |
| Spare-time activity[a] | 2.1±0.7 | 2.1±0.6 | 2.1±0.8 | −0.608 | 0.544 | 2.1±0.8 | 2.2±0.8 | −1.756 | 0.084 |
| Before-school activity | 2.1±1.3 | 1.9±1.4 | 2.2±1.3 | −1.076 | 0.283 | 2.1±1.5 | 2.0±1.3 | 1.033 | 0.305 |
| Physical Education | 3.9±1.1 | 4.0±1.1 | 3.9±1.1 | 0.875 | 0.383 | 3.7±1.3 | 3.5±1.2 | 1.930 | 0.058 |
| Recesses | 2.9±1.0 | 2.7±0.9 | 3.1±1.0 | −2.260 | 0.025* | 2.6±0.9 | 2.7±1.0 | −0.685 | 0.495 |
| After school | 3.0±1.1 | 3.1±1.1 | 3.0±1.2 | 0.434 | 0.665 | 3.0±1.1 | 3.0±1.0 | 0.281 | 0.780 |
| Evenings | 2.6±1.0 | 2.6±1.0 | 2.6±1.1 | −0.277 | 0.782 | 2.5±1.2 | 2.6±1.1 | −0.617 | 0.539 |
| Weekend | 2.9±1.0 | 3.0±0.8 | 2.9±1.1 | 0.201 | 0.841 | 2.9±1.0 | 2.8±0.9 | 0.820 | 0.415 |
| Statement | 2.8±1.1 | 2.7±1.1 | 2.8±1.0 | −1.073 | 0.285 | 2.7±1.0 | 2.7±1.0 | −0.155 | 0.877 |
| Weekly activity | 3.0±0.8 | 3.1±0.7 | 2.9±0.8 | 1.315 | 0.190 | 2.8±0.7 | 2.6±0.8 | 1.875 | 0.066 |
| **Accelerometer-determined PA (n = 117; 57 girls)** | | | | | | | | | |
| Total time of PA (min) | 278.5±49.2 | 271.1±44.4 | 285.5±52.8 | −1.601 | 0.112 | – | – | – | – |
| Total time of MVPA (min) | 52.8±17.3 | 51.7±16.2 | 53.8±18.4 | −0.655 | 0.514 | – | – | – | – |
| Percent day MVPA | 6.6±2.1 | 6.4±2.1 | 6.8±2.1 | −1.001 | 0.319 | – | – | – | – |

[a]Spare-time activity item (Q1) rescaled = 4(Q1−1) / [(sample max raw score)− 1] + 1, where sample max raw scores were 2.23, 2.18, and 1.95 for the measure in phase one, the 1st measure in phase two, and the 2nd measure in phase two, respectively.

MVPA: moderate to vigorous physical activity; PA: physical activity; PAQ-C/CZ: the Czech version of Physical Activity Questionnaire for Older Children.

*p < 0.05 denotes statistically significant difference.

overall gender effect (Wilks-Lambda: $F_{(9, 159)} = 1.448$; $p = 0.156$; $\eta^2 = 0.08$). Independent sample T-test identified significant gender differences in recesses ($t(167) = 2.260$; $p = 0.025$; $d = 0.35$) and BMI ($t(167) = -2.982$; $p = 0.003$; $d = 0.45$).

## Internal consistency and item/scale relationship

Standardized Cronbach's alpha was 0.77 (95% CI: 0.70–0.82), which is acceptable for general research purposes [39]. CITCs values differed widely across the items and ranged from 0.29 to 0.61 (Table 2) indicating a homogeneous scale [40]. The results provide evidence that the PAQ-C/CZ has acceptable item/scale properties and good scale reliability in this population [39].

## Exploratory factor analysis

In order to identify the factor structure of the PAQ-C/CZ, an EFA employing the Maximum Likelihood method was performed on data. Kaiser-Meyer-Olkin test indicated the sample was large enough (measure of sampling adequacy >0.80). Based on Kaiser's criterion, we identified one factor only with eigenvalue greater than one (the two highest eigenvalues were 3.33 and 0.997). The factor explains 37.00% of variance, which is not satisfactory. The factor load varied

**Table 2. Psychometric properties for the PAQ-C/CZ: Item/scale relationship (CITCs), factor loadings, concurrent validity (Spearman's r), and test-retest reliability (ICC).**

| | Phase one | | | Phase two |
|---|---|---|---|---|
| | CITCs (n = 169) | Factor load (n = 169) | Spearman's r [b] (n = 117) | ICC [b] (n = 63) |
| PAQ-C/CZ total score[a] | – | – | 0.28** | 0.94** |
| | | | (0.12–0.43) | (0.90–0.96) |
| Spare-time activity[a] | 0.35 | 0.42 | 0.21* | 0.92** |
| | | | (0.07–0.36) | (0.87–0.95) |
| Before-school activity | 0.32 | 0.33 | 0.05 | 0.86** |
| | | | (−0.10–0.20) | (0.77–0.92) |
| Physical Education | 0.29 | 0.32 | 0.14 | 0.92** |
| | | | (−0.02–0.30) | (0.87–0.95) |
| Recesses | 0.32 | 0.33 | 0.19* | 0.91** |
| | | | (0.04–0.34) | (0.85–0.94) |
| After school | 0.61 | 0.69 | 0.14 | 0.78** |
| | | | (−0.02–0.29) | (0.63–0.87) |
| Evenings | 0.45 | 0.50 | 0.19* | 0.73** |
| | | | (0.03–0.34) | (0.56–0.84) |
| Weekend | 0.58 | 0.72 | 0.12 | 0.81** |
| | | | (−0.04–0.28) | (0.68–0.88) |
| Statement | 0.55 | 0.67 | 0.19* | 0.81** |
| | | | (0.05–0.33) | (0.68–0.88) |
| Weekly activity | 0.59 | 0.73 | 0.26** | 0.85** |
| | | | (0.11–0.42) | (0.75–0.91) |

[a] Spare-time activity item (Q1) rescaled = 4(Q1–1) / [(sample max raw score)– 1] + 1, where sample max raw scores were 2.23, 2.18, and 1.95 for the measure in phase one, the 1st measure in phase two, and the 2nd measure in phase two, respectively.

[b] The 95% confidence intervals are in parentheses.

CITCs: corrected item-total correlations; CI: confidence interval; ICC: intraclass correlation coefficient; PAQ-C/CZ: the Czech version of Physical Activity Questionnaire for Older Children.

* $p < 0.05$;

** $p < 0.01$.

from 0.32 to 0.73 (Table 2), suggesting a heterogeneous factor load and a questionable factor structure. The three lowest loads were recorded in the items related to in-school activities, i.e. *recesses* and *PE*, and in the item *before-school activity*. The other loads were higher than 0.42.

## Test-retest reliability

The ICCs ranged from 0.73 to 0.94 (Table 2) and they were statistically significant (p < 0.001). The highest ICC value was observed in the PAQ-C/CZ total score. The test-retest differences in both the PAQ-C/CZ total and PAQ/CZ items' scores were negligible (ranged from –0.1 to 0.2) and not statistically significant (p > 0.05) except the spare-time activity item in raw score (p = 0.014, d = 0.32) (Table 1). The results suggest good test-retest reliability of PAQ-C/CZ.

## Concurrent validity

The measurement of concurrent validity was based on correlation (Spearman's r) between PAQ-C/CZ score and the percentage of time spent in MVPA assessed by accelerometer. PAQ-C/CZ total score correlated significantly with the MVPA (r = 0.28; 95% CI: 0.12–0.43). The correlations varied in particular items (r = 0.05–0.26; six out of nine were statistically significant) (Table 2). All observed values point out a low association between PAQ-C/CZ data and objectively assessed MVPA. This indicates lower concurrent validity than would be expected.

## Discussion

This study presents the third validated questionnaire for the assessment of PA applicable for use in the Czech population of older children. The PAQ-C/CZ varies from the other two questionnaires adapted into the Czech language (Health Behaviour in School-aged Children Questionnaire and Youth Activity Profile questionnaires) in several aspects [10–13]. First, the PAQ-C/CZ is considerably shorter, less difficult, requires less time, and therefore the children can concentrate when fulfilling the questionnaire. Second, the questionnaires' outputs also differ. Unlike the PAQ-C/CZ, the first-mentioned questionnaire is focused on a comprehensive assessment of healthy behaviour and the second one on the estimation of values comparable to the objective PA measurement. As such, the questionnaire expands the current possibilities of PA assessment. This study depicts all aspects of the adaptation of the PAQ-C into the Czech language and presents relevant psychometric properties.

The questionnaire translation respected all general recommendations [28], and cultural differences were taken into consideration. This led to our modifying the *spare-time activities* checklist as we mentioned in the Methods section to better reflect the structure of school time in Czechia and children's overall movement behaviour during school days. Such changes and replacements to such questionnaires are not isolated—similar item modifications, removals, or additions were made in other language versions or modifications of the PAQ-C as well [17–19, 21, 22, 24, 25, 44, 45]. Data collection was practical in that completion of the questionnaire took 10–35 minutes, which is acceptable. In accordance with Janz et al. [32], the individual raw score in the *spare-time activity* item was rescaled to reflect a range consistent with the other questions. In the study, the highest score was observed in the *Physical Education* item. This finding is consistent with other studies evaluating the PAQ-C properties [29]. It can be assumed that it is related to feedback about the level of student activity which is customarily provided by a PE teacher. In contrast to the study by Gobbi et al. [21], no overall gender effect was observed in our case. Considering particular items, gender effect was registered only in *recesses* where boys scored significantly higher than girls (p < 0.05). That is not fully in agreement with the findings in the PAQ-C original version where boys scored significantly higher

in all the items and the total score in comparison to girls [29]. In the Czech Republic, there has been no evidence of objectively evaluated PA based on systematic research in this age group so far. Therefore, it is virtually impossible to provide further interpretation of the findings. For instance, it is not possible to determine whether they are specific for Czech children of this age in general or whether the actual recruitment of participants caused a distortion. That said, the findings present a welcome challenge to conduct further research.

The results of Cronbach's alpha (0.77) and CITCs (range between 0.29–0.61) indicated satisfactory internal consistency and acceptable item/scale properties of the PAQ-C/CZ [40]. However, the wide range of observed CITCs shows the scale is not fully homogeneous. Especially, the values in the first four items indicate that they contribute less to the total PAQ-C/CZ score. Nevertheless, the observed levels of internal consistency and item/scale properties are similar to the original Canadian (alpha = 0.70–0.83; CITCs were not investigated) [46] and other modified versions of the PAQ-C (alpha = 0.77–0.83; CITCs range from the lowest limit 0.04 to the highest limit 0.99) [15, 17, 19, 21, 25, 47–49].

The results of EFA suggested that the PAQ-C/CZ is a one-dimensional structure, where one factor mostly measures one construct, presumably MVPA. The observed model showed factor loads ranging widely from 0.32 to 0.73, whereas items *before-school activity*, *physical education*, and *recesses* only contribute to the total score a little. This probably corresponds to low variability explained by the factor (37.0%) and suggests there are still many other latent variables explaining the total score in the questionnaire. Factor load in four items is under 0.50, which is low. Similar to other studies determining the one-factor structure of the PAQ-C, the highest factor loadings were observed in *weekly activity*, *weekend*, *after school*, and *statement* [17, 29, 32]. However, there are studies reporting two-factor structure, in which one factor is frequently composed of items assessing *in-school activities* with all others pertaining to the second one [19, 21, 25, 48]. To the best of our knowledge, there is only one study in which a three-factor structure was reported [47]. In fact, the EFA results are not outright contrary to the mentioned two-factor structure. In accordance with that structure, the lowest factor load was also found in the item *before school* and those items related to school (*recesses* and *Physical Education*); this could indicate the presence of another factor related to these items. Although the tendency of the items' factor load is close to those presented in the mentioned studies, factor loads in the first four items are under 0.50, which is low. Unfortunately, the data did not provide evidence of another factor that would better describe the structure of the questionnaire. It would be desirable for further research to suggest how to strengthen the factor structure.

In this study, the concept of the assessment of the test-retest reliability differs from other studies determining the psychometric properties of PAQ-C. For the purpose of an assessment of test-retest reliability, they frequently use a design based on completing the questionnaire, for instance of one-week apart or longer and each time recalling physical activity in two different 7-day periods. In such a way, some bias of reliability assessment can be caused by the variability of an observed phenomenon (here physical activity). The approach taken in this study should eliminate a bias related to this effect. This approach is designed in such a way that the participant twice recalls physical activity in the same time period in two separate days. The question remains what the optimum distance between the repeated completion of the questionnaire should be. The advantage of a minimum time gap is that pupils must be able to recall PA conducted in the last seven days. On the other hand, the respondent should not be influenced by the questionnaire's previous completion; thus, a broader time gap may be preferable. In consideration of the participants' age, we opted for the distance of one day in our study.

Therefore, the participants completed the questionnaire concerning their PA twice within 2 subsequent days during the identical 7-day period. Considering this approach, most discrepancies between the repeatedly obtained data would be explained by a low quality of the

questionnaire (including the whole protocol of data collection) or by the effort the children would make to fill in the answers identically when completing the questionnaire for the second time. However, we consider this approach more appropriate for a test-retest reliability assessment. In both approaches, the results should be interpreted with caution. The test-retest reliability was investigated employing ICC and the Student's paired T-test. It was determined for the total as well as items' scores. The ICCs showed good to excellent reliability [43]. The highest value was observed in the total score (ICC = 0.94; 95% CI: 0.90–0.96). Concurrently, all test-retest differences in the total and items' scores were non-significant and ranged from –0.1 to 0.2, which may be considered negligible. The findings demonstrate a good 1-day reproducibility.

The association between the PAQ-C/CZ and accelerometer-determined MVPA was measured to assess the concurrent validity. Statistically significant correlation was observed between the percentage of time spent in MVPA and the PAQ-C/CZ total score (Spearman's r = 0.28, 95% CI: 0.12–0.43). The value is lower in comparison to the study reporting concurrent validity of the original PAQ-C version (r = 0.39) in which Caltrac accelerometer was used as a concurrent method [29]. Surprisingly, Janz et al. [32] reported a much higher value (r = 0.63) in the subsequent validation of the same version of the PAQ-C. Although they employed the ActiGraph wGT3X-BT accelerometer as a concurrent method, they applied other cut-off points to determine MVPA, which can lead to distinct quantification of MVPA. However, the association between the PAQ-C and accelerometer data observed in this study is close to the findings from other studies (0.12 to 0.47) which validated various language adaptations of the PAQ-C followed similar protocols [16, 17, 21, 44, 45, 48]. The variability of correlations between the PAQ-C/CZ score in particular items and accelerometer data showed different capabilities of the items to assess MVPA. Based on the observed correlations, the highest and lowest capabilities can be expected in the *weekly* and *before-school activity* items, respectively. Considering all the observed values of correlations, it can be concluded the ability of the PAQ-C/CZ to estimate MVPA is quite low in comparison to the objective technique based on an accelerometry as well as to the original version of the questionnaire. The PAQ-C should primarily assess MVPA [26]. However, given the EFA and concurrent validity results, the question arises as to whether the set of items describe such a construct. It is worth mentioning here that the nature of the assessments of PA employing self-reporting questionnaires on the one hand and accelerometers on the other were distinct. Therefore, any assumption of an excellent correlation between MVPA determined by an accelerometer and a questionnaire would be faulty. This is in sync with findings reported in other studies evaluating concurrent validity of other known questionnaires assessing PA in the population of children. No reported correlations (with comparative method) exceeded 0.55 [50]. In general, the questionnaires assessing PA were more comprehensive, studying it from more perspectives. The time spent in PA with a particular intensity tends to be only one of explicative PA dimensions. Considering these facts, the study results indicate that the PAQ-C/CZ is a tool for the assessment of PA in Czech older children with limited capability to assess MVPA. Therefore, emphasis must be placed on appropriate and careful results interpretation to prevent bias. However, it is necessary to assess the validity of the PAQ-C/CZ in relation to other aspects of PA (the type and frequency of PA, sociological aspects, environmental aspects, fitness level, etc.) [51, 52].

Due to the PAQ-C/CZ properties and psychometric quality comparable to the other European PAQ-C versions, the PAQ-C/CZ could be recommended for the purposes of PA assessment in any national or European studies. However, the results of any comparative study need to be carefully interpreted because of some content differences across various language modifications of the PAQ-C. A self-reported questionnaire is a very specific concept of the assessment of movement behaviour. Therefore, if possible, it would be beneficial to also employ a PA assessment tool evaluating other aspects to draw a comprehensive description of PA.

## Strengths and limitations

This study has several strengths. First, the questionnaire modifications related to socio-cultural specifics were supported by previous studies related to movement behaviour in children in the Czech Republic [30, 31]. Second, the results are comparable with those reported in the majority of recent studies validating PAQ-C which followed similar protocols.

On the other hand, some limitations have to be mentioned. A non-randomised sampling method was used, which means the findings are not fully generalisable. Moreover, the next non-observed variables—such as the level of reading and text comprehension, the ability to recall activities in the past, understanding the PA intensity levels, or social desirability—could be considered potential sources of error [18, 29]. The effects of these variables on psychometric properties still need to be investigated in future studies. Although the PAQ-C is designed for children aged 8–14 years, this study involved participants whose ages ranged between 10–13 years. This limited the generalisability of the results for younger as well as older children, together with the comparability of results to other studies validating the PAQ-C. Nevertheless, the elementary school teachers cooperating in the study pointed out that the demands the PAQ-C/CZ posed on reading and comprehension made this questionnaire inappropriate for children below 10 years of age. Taking that into consideration, it was consequently decided that the PAQ-C/CZ will only be used with children 10+ years old. Moreover, 14-year-olds were not included in the study. When selecting the sample, we estimated that if the questionnaire was comprehensible to the children aged 10–13 years, older pupils would find it understandable as well. Finally, not all participants adhered to the accelerometery wear protocol. Although there is no evidence for systematic bias related to participant characteristics (demographical, movement behavioural, or PAQ-C score), it is possible that the more active the child, the more likely s/he was to wear the accelerometer. Therefore, it is up for discussion whether the reason why children have no valid data can somehow be related to the validity or reliability the PAQ-C/CZ.

## Conclusions

In the study, the first adaptation of the PAQ-C into the Czech language was described (The Czech version of the questionnaire is available in the S1 File). The psychometric properties of the questionnaire were evaluated, including internal consistency, item/scale relationship, factor structure, test-retest reliability, and concurrent validity. The findings suggested the PAQ-C/CZ is an instrument measuring one construct with a moderate internal consistency, homogenous scale, and good reliability in Czech children (10–13 years old). Borderline concurrent validity indicated that the PAQ-C/CZ can assess MVPA with certain limitations in children. Considering all the observed psychometric properties, it is recommended to use PAQ-C/CZ in large-sample research studies only and with emphasis on a correct interpretation of results. Given the limitations mentioned above, further research is required in order to improve the psychometric properties of the questionnaire, and to make further adjustments to it if necessary. It is especially vital to explain the questionnaire's validity from a more general perspective, to analyse gender effect in all psychometric properties, and to analyse sources of error.

## Supporting information

**S1 Data. Czech adaptation of PAQ-C.**
(TXT)

**S1 File. The comparison of PAQ-C and PAQ-C/CZ.**
(DOCX)

**S2 File. The Physical Activity Questionnaire for Older Children—The Czech version.**
(PDF)

## Acknowledgments

The authors would like to thank all school directors, children, and their parents who agreed to participate in this study. Special thanks as well to the expert panel and school teachers who evaluated the content and semantic aspects of PAQ-C/CZ. Finally, the authors very much appreciate the cooperation of the Ph.D. students who participated in data collection.

## Author Contributions

**Conceptualization:** Roman Cuberek, Marcela Janíková.

**Data curation:** Roman Cuberek.

**Formal analysis:** Roman Cuberek, Jan Dygrýn.

**Investigation:** Roman Cuberek, Marcela Janíková.

**Methodology:** Roman Cuberek, Jan Dygrýn.

**Project administration:** Marcela Janíková.

**Resources:** Marcela Janíková.

**Software:** Roman Cuberek, Jan Dygrýn.

**Supervision:** Roman Cuberek.

**Writing – original draft:** Roman Cuberek, Marcela Janíková.

**Writing – review & editing:** Roman Cuberek, Marcela Janíková, Jan Dygrýn.

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
