## [Decision Letter · Decision Letter 0]

20 Oct 2020

PONE-D-20-26808

Adaptation and validation of the Physical Activity Questionnaire for Older Children among Czech children

PLOS ONE

Dear Dr. Cuberek,

Thank you for submitting your manuscript to PLOS ONE. After careful consideration, we feel that it has merit but does not fully meet PLOS ONE’s publication criteria as it currently stands. Therefore, we invite you to submit a revised version of the manuscript that addresses the points raised during the review process.

The manuscript has been reviewed by two experts who both have made important comments that should be addressed. Based on these comments and my own reading of the manuscript, I strongly recommend the authors to take a more critical and less optimistic stance on the findings of the actual psychometric properties of the PAQ-C throughout the abstract, results and conclusions. For instance, as reviewer 2 correctly notes, Cronbach’s alpha value points at acceptable internal consistency (for research purposes only), item-total correlations differ widely, and the correlation with the accelerometer data is very poor for establishing “concurrent validity” to all common measurement standards (and also according to your own cutoffs as described in the analysis). This should be better reflected in the now very positively framed conclusions in the abstract and more extensively discussed in the introduction. Even if these findings are in line with previous PAQ-C validation studies, this does have implications for the validity and applicability of the instrument (or self-reported PA instruments in general) for different purposes and/or populations.

Also, the results with respect to the number of factors underlying the instrument are a bit puzzling to me. Parallel analysis and Kaiser's eigenvalue greater than 1.0 rule are two different methods of determining the optimal number of factors underlying a set of items. Since parallel analysis isn’t based on the number of factors with “… eigenvalue >1 … (page 11, line 212), this is either reported or performed incorrectly (I assume the first, since parallel analysis isn’t standard included in SPSS). The authors should clearly describe how they determined the number of factors underlying the instrument in the statistical analysis section with the appropriate references (not in the results section) and the actual eigenvalues of at least the first two factors should be presented in the results. Also, the rather low explained variance of this factor (usually we aim for around 50%) should be reflected on in the discussion.

Finally, as the authors mention themselves, they took a rather different approach for examining test-retest reliability by having children report on the same recall days on two sperate days. Please comment in the discussion on why this was done this way and if this can be meaningfully comparted with the usual practice of completing an instrument for instance one-week apart.

We look forward to receiving your revised manuscript.

Kind regards,

Peter M ten Klooster, Ph.D.

Academic Editor

PLOS ONE

Journal Requirements:

2) Please amend either the title on the online submission form (via Edit Submission) or the title in the manuscript so that they are identical.

Reviewers' comments:

Reviewer's Responses to Questions

**Comments to the Author**

1. Is the manuscript technically sound, and do the data support the conclusions?

Reviewer #1: Yes

Reviewer #2: Partly

2. Has the statistical analysis been performed appropriately and rigorously? 

Reviewer #1: Yes

Reviewer #2: No

3. Have the authors made all data underlying the findings in their manuscript fully available?

Reviewer #1: Yes

Reviewer #2: Yes

4. Is the manuscript presented in an intelligible fashion and written in standard English?

Reviewer #1: No

Reviewer #2: Yes

5. Review Comments to the Author

Reviewer #1: General Comments:

Overall, this paper uses strong methodology, which has previously been used in several other validation studies, to validate the PAQ-C in Czech children. The modifications of the questionnaire enable it to be culturally specific and the concurrent validity and intra-class correlations are similar to those from other countries demonstrating that this is an acceptable questionnaire to measure MVPA in this population.

However, I am interested in the generalizability of this population, as I find this the main limitation of the paper. The children were from one area of Czech Republic and only span the ages of 10-13. The proportion of the population that this questionnaire benefits is therefore potentially small. It would be useful to have the authors discuss the socio-economic status of the region and the range of schools in the study to further understand the range of children involved.

The paper would benefit from English editing throughout to correct grammatical mistakes. They are present on all pages of manuscript. For example, on page 3, line 50: the word “because” is not necessary; line 57: requires rewording; line 61: “questionnaires in Czech-language modifications” should be “with modifications”; line 66: “have” should be “has” as you are referring to the PAQ-C.

Specific Comments:

Introduction

Page 3, line 46-49: Whilst COVID is at the forefront of the news and current discussions, I do not believe that this paper benefits from referencing the virus. If the authors wish to keep this, I recommend focusing on how older children’s activity patterns may have changed due to various restrictions and not on how promoting PA could strengthen the immune system. However, I would recommend expanding on the first two sentences of the introduction instead.

Page 3, line 51: “movement behaviour” – I do not believe that the word movement is necessary. It could also be moved further up in the sentence “Measuring PA is a challenging task as movement is a complex, multi-dimensional human behaviour”.

Page 3, line 63 -65: A citation is necessary for these claims

Methods

Page 5, line 114: Could the authors please tell the readers more about the schools and location that the test was administered? Is this area and the schools generalizable to the rest of Czech Republic?

Page 5, line 119 - 120: Could the authors please state why the expert board and teachers felt that students under the age of 10 would be unable to complete the questionnaire accurately? Is there a reason that children aged 14 were not invited to participate in the study? My understanding is that the Canadian version was intended for students in grades 4 – 8, who could be between the ages of 8 – 14 years.

Results

Page 7, lines 177 – 179: Are there students that participated in both phases of the study as 169 + 63 takes me above the 223 children stated in line 177. Were the 14 excluded subjects from both phase 1 and 2? If the students that did the accelerometry in phase 1 are different from those who did re-test reliability in phase 2, I think this must be made clear in the methodology and the reasoning explained.

Page 11, line 221: not statistically significant needs to replace statistically non-significant. They imply different things. We cannot claim that it is non-significant, only that it isn’t significant.

Table 1

Phase 2, n does not include the number of girls

Table 2

Phase 2 includes concurrent validity (the 139 students that did both the accelerometry data and PAQ-C). According to the methods section on page 6, this is part of phase 1.

Discussion

Page 11, line 232: How does this questionnaire differ from the other 2 stated questionnaires? Is it shorter? Does it require less time? Is it more or less difficult? Please state the direction as readers may not be familiar with the other two questionnaires.

Page 12, line 251 – 253: Do the authors have any thoughts as to why there was no difference between boys and girls in this population? This was also shown in your accelerometry data for MVPA and could add to the discussion.

Page 15, line 320: I would caution the authors on using the words “identical methods” as it implies all the details are the same, which is not true. For example, the Polish version and the Chinese version used 5s epochs to quantify MVPA by accelerometry, which differs to the 15s used here. I would recommend “followed similar protocols”.

Page 15, line 326 – 332: Here the authors explain why children below 10 were not used in the test, but is there a reason that 14-year-old children were also excluded? Could the authors please explain.

Conclusion

Page 15, line 341: The first 4 words of the sentence do not flow into the rest of the sentence.

Reviewer #2: Below are my comments:

1. The results of the PAQ-C/CZ indicated satisfactory internal consistency (Cronbach’s alpha = 0.77), not good.

2. The item/scale properties indicated diverse item-total correlations (corrected item-total correlations ranged between 0.29–0.61). It needs more explanations and justifications.

3. The factor loading of the exploratory factor analysis ranging from 0.30 to 0.79 which can't demonstrate the acceptance level in factor analysis for this study.

4. The concurrent validity between the PAQ-C/CZ total score and the accelerometer-determined MVPA is 0.29 which is quite low and can't reflect the objective PA level by the self-reported questionnaire through this study.

6. PLOS authors have the option to publish the peer review history of their article (what does this mean?). If published, this will include your full peer review and any attached files.

Reviewer #1: **Yes: **Stephanie Duncombe

Reviewer #2: No

---

## [Author Response · Author response to Decision Letter 0]

7 Dec 2020

RESPONSES TO ACADEMIC EDITOR

Commentary #1: 

Cronbach’s alpha value points at acceptable internal consistency (for research purposes only), item-total correlations differ widely, and the correlation with the accelerometer data is very poor for establishing “concurrent validity” to all common measurement standards (and also according to your own cutoffs as described in the analysis).

We have taken the comment into account and we did several changes in the manuscript:

1. Page 10, lines 208 – 209 (Results): 

We corrected the sentence “Standardized Cronbach’s alpha was 0.77 (95% CI: 0.70–0.82), which is acceptable.” by „Standardized Cronbach’s alpha was 0.77 (95% CI: 0.70–0.82), which is acceptable for general purposes.” Furthermore, we added a sentence “CITCs values differed widely across the items and ranged from 0.29 to 0.61.”

2. Page 11, lines 243 – 246 (Results): 

We replaced two sentences “This supports the assumption that the questionnaire can assess MVPA within a week. The correlations varied in particular items (r = 0.05–0.26; six out of nine were statistically significant) (Table 2).” with “The correlations varied in particular items (r = 0.05–0.26; six out of nine were statistically significant) (Table 2). All observed values point out a low association between PAQ-C/CZ data and objectively assessed MVPA. This indicates lower concurrent validity than would be expected.”

3. Page 15, lines 341 – 345 (Discussion):

In order to better interpret the results, we have expanded the section on concurrent validity by three sentences placed after the sentence “Based on the observed correlations, the highest and lowest capabilities can be expected in the weekly and before-school activity items, respectively.” 

They are: “Considering all the observed values of correlations, it can be concluded the ability of the PAQ-C/CZ to estimate MVPA is quite low in comparison to the objective technique based on an accelerometry as well as to the original version of the questionnaire. The PAQ-C should primarily assess MVPA. However, given the EFA and concurrent validity results, the question arises as to whether the set of items describe such a construct.”

4. Page 15, lines 353 – 355 (Discussion):

In the same order two sentences were replaced: “Considering these facts, the study results indicate that the PAQ-C/CZ is a sufficiently valid tool for the assessment of habitual MVPA in Czech older children. The PAQ-C can be recommended as an appropriate instrument for the Czech Ministry of Education, Youth and Sports to collect evidence of the level of PA in Czech children as detailed in the Introduction.” with “Considering these facts, the study results indicate that the PAQ-C/CZ is a tool for assessment of PA in Czech older children with limited capability to assess MVPA. Therefore, emphasis must be placed on appropriate and careful results interpretation to prevent bias.”

Commentary #2: 

This should be better reflected in the now very positively framed conclusions in the abstract and more extensively discussed in the introduction. Even if these findings are in line with previous PAQ-C validation studies, this does have implications for the validity and applicability of the instrument (or self-reported PA instruments in general) for different purposes and/or populations.

The commentary was taken into consideration in Conclusions and Abstract, which was rewritten and reflected the changes accomplished in the whole paper based on commentary of the Academic Editor and both Reviewers as well.

1. Page 17, line 396 (Conclusions): 

We replaced part of the sentence “…a moderate to good…” with “…a moderate…”. 

2. Page 17, lines 397 – 401 (Conclusions):

In accordance with all previous corrections and changes two sentences were replaced in this section. 

“Concurrent validity analysis indicated that the PAQ-C/CZ could assess MVPA with acceptable validity. This instrument could also be used in future large-sample studies. However, given the limitations mentioned above, further research is necessary.” was replaced with “Borderline concurrent validity indicated that the PAQ-C/CZ can assess MVPA with certain limitations in children. Considering all the observed psychometric properties, it is recommended to use PAQ-C/CZ in large-sample research studies only and with emphasis on a correct interpretation of results. Given the limitations mentioned above, further research is required in order to improve the psychometric properties of the questionnaire, and to make further adjustments to it if necessary.”. 

Commentary #3: 

Also, the results with respect to the number of factors underlying the instrument are a bit puzzling to me. Parallel analysis and Kaiser's eigenvalue greater than 1.0 rule are two different methods of determining the optimal number of factors underlying a set of items. Since parallel analysis isn’t based on the number of factors with “… eigenvalue >1 … (page 11, line 212), this is either reported or performed incorrectly (I assume the first, since parallel analysis isn’t standard included in SPSS). The authors should clearly describe how they determined the number of factors underlying the instrument in the statistical analysis section with the appropriate references (not in the results section) and the actual eigenvalues of at least the first two factors should be presented in the results. Also, the rather low explained variance of this factor (usually we aim for around 50%) should be reflected in the discussion.

1. Page 7, lines 175 – 176 (Methods):

We added a new sentence “Kaiser’s criterion was used to determine the number of factors to be retained (eigenvalue greater than one).”

2. Page 11, lines 224 – 231 (Results):

We replaced the paragraph “In order to establish the construct validity…” with “In order to identify the factor structure of the PAQ-C/CZ, an EFA employing Maximum Likelihood method was performed on data. Kaiser-Meyer-Olkin test indicated the sample was large enough (measure of sampling adequacy >0.80). Based on Kaiser’s criterion, we identified one factor only with eigenvalue greater than one (the two highest eigenvalues were 3.33 and 0.997). The factor explains 37.00% of variance, which is not satisfactory. The factor load varied from 0.32 to 0.73 (Table 2), suggesting a heterogeneous factor load and a questionable factor structure. The three lowest loads were recorded in the items related to in-school activities, i.e. recesses and PE, and in the item before-school activity. The other loads were higher than 0.42.” 

3. Page 13, lines 288 – 291 (Discussion): 

We extended the sentence “The observed model showed factor loads ranging from 0.32 to 0.73.” with “…, whereas items before-school activity, physical education, and recesses only contribute to the total score a little. This probably corresponds to low variability explained by the factor (37.0%) and suggests there are still many other latent variables explaining the total score in the questionnaire. Factor load in four items is under 0.50, which is low.”

Additional commentary: 

It was found that extending the analysis by another factor (its eigenvalue = 0.997) will increase the explained variability up to 48.1%. However, the subsequent analysis did not prove to be beneficial in terms of the identified factor loads. 

We believe that this information will be helpful to clarify our response. However, we do not think it is appropriate to include them in the results section.

Commentary #4: 

Finally, as the authors mention themselves, they took a rather different approach for examining test-retest reliability by having children report on the same recall days on two separate days. Please comment in the discussion on why this was done this way and if this can be meaningfully compared with the usual practice of completing an instrument for instance one-week apart.

1. Page 13 and 14, lines 305 – 315 (Discussion): 

Considering the commentary, we added this information: “For the purpose of an assessment of test-retest reliability, they frequently use a design based on completing the questionnaire, for instance of one-week apart or longer and each time recalling physical activity in two different 7-day periods. In such a way, some bias of reliability assessment can be caused by the variability of an observed phenomenon (here physical activity). The approach taken in this study should eliminate a bias related to this effect. This approach is designed in such a way that the participant twice recalls physical activity in the same time period in two separate days. The question remains what the optimum distance between the repeated completion of the questionnaire should be. The advantage of a minimum time gap is that pupils must be able to recall PA conducted in the last seven days. On the other hand, the respondent should not be influenced by the questionnaire's previous completion; thus, a broader time gap may be preferable. In consideration of the participants' age, we opted for the distance of one day in our study.“ As a follow-up to this modification we deleted one sentence “The study focused on the capability of the questionnaire to repeatedly produce the same results about children's PA when the same 7-day behaviour is assessed.”

2. Page 14, lines 320 – 322 (Discussion): 

In accordance with the changes above, we added a new sentence “However, we consider this approach more appropriate for a test-retest reliability assessment. In both approaches, the results should be interpreted with caution.”

RESPONSES TO REVIEWER #1:

Commentary #1: I am interested in the generalizability of this population, as I find this the main limitation of the paper. The children were from one area of Czech Republic and only span the ages of 10-13. The proportion of the population that this questionnaire benefits is therefore potentially small. It would be useful to have the authors discuss the socio-economic status of the region and the range of schools in the study to further understand the range of children involved.

The commentary is reflected on page 5, lines 115 – 122:

We added new sentences to explain the socio-economic status of the Czech Republic and to understand the sampling/range of children involved:

“The Czech Republic is a rather small country with a population of approx. 10 million, a unified educational system, and a social-economic status comparable across the population. The Czech Republic is a relatively small country with a population of approx. 10 million, a unified educational system, and a social-economic status comparable across the population. In selecting the sample of schools, the municipality's size (large, medium, or small) was taken into account as this is a factor that corresponds with the availability and opportunity of partaking in various types of PA. It also reflects the diverse social-economic environments. Regarding the study's aim, we consider such a sample adequately representative of the country's school-children and youth population.”

Commentary #2: The paper would benefit from English editing throughout to correct grammatical mistakes. They are present on all pages of the manuscript. For example, on page 3, line 50: the word “because” is not necessary; line 57: requires rewording; line 61: “questionnaires in Czech-language modifications” should be “with modifications”; line 66: “have” should be “has” as you are referring to the PAQ-C.

Language editing of the paper, before submitting, was provided by Elsevier agency (British English) as recommended in guidelines for authors. 

1. Page 3, line 51: We removed the word “because”.

2. Page 3, lines 58 – 59: We reworded the sentence “In the last 30 years, national data on PA and physical fitness of Czech children are rare or subpar .” by “In the last 30 years, national data on PA and physical fitness of Czech children have been hardly comprehensive.”

3. Page 3, line 62: We replaced “Czech-language modifications” with “Czech-language with modifications”.

4. Page 3, line 67: We replaced “have” with “has”.

Commentary #3 (Introduction): Page 3, lines 46-49: Whilst COVID is at the forefront of the news and current discussions, I do not believe that this paper benefits from referencing the virus. If the authors wish to keep this, I recommend focusing on how older children’s activity patterns may have changed due to various restrictions and not on how promoting PA could strengthen the immune system. However, I would recommend expanding on the first two sentences of the introduction instead.

Page 3, lines 42 – 50: 

The first paragraph was replaced with “Due to an ample amount of research into physical activity (PA) in school-aged children and youth in recent years, the benefits of PA are well-documented. The key finding is that movement behaviour at this age influences the behaviour in adulthood considerably [1–3]. What transpires is that PA brings numerous benefits to school-aged children and youth, such as improved cardiorespiratory fitness, muscular strength, and mental health, and lowers the risk of overweight, obesity, and metabolic syndrome related to other diseases [4]. It is considered a critical supportive factor in developing cognitive function, positively impacting academic achievement besides [5]. The aforementioned benefits of PA are crucial for creating and evaluating the government's intervention programmes aimed at educational and health areas. This requires regular monitoring of the current state of PA in school-aged children and youth.” 

Commentary #4 (Introduction): Page 3, line 51: “movement behaviour” – I do not believe that the word movement is necessary. It could also be moved further up in the sentence “Measuring PA is a challenging task as movement is a complex, multi-dimensional human behaviour”.

Page 3, lines 63 -65: A citation is necessary for these claims

Page 3, lines 51 – 52: 

We replaced the sentence “Measuring PA is a challenging task as it is a complex, multi-dimensional human movement behaviour, warranting a multi-data approach.” with “Measuring PA is a challenging task as movement is a complex, multi-dimensional human behaviour, warranting a multi-data approach.” 

Commentary #5 (Introduction): Page 3, lines 63 -65: A citation is necessary for these claims.

Page 3, line 66: Citations were added.

Commentary #6 (Methods): Page 5, line 114: Could the authors please tell the readers more about the schools and location that the test was administered? Are this area and the schools generalizable to the rest of the Czech Republic?

Page 5, lines 115 – 122: 

Four sentences were added: “The Czech Republic is a rather small country with a population of approx. 10 million, a unified educational system, and a social-economic status comparable across the population. The Czech Republic is a relatively small country with a population of approx. 10 million, a unified educational system, and a social-economic status comparable across the population. In selecting the sample of schools, the municipality's size (large, medium, or small) was taken into account as this is a factor that corresponds with the availability and opportunity of partaking in various types of PA. It also reflects the diverse social-economic environments. Regarding the study's aim, we consider such a sample adequately representative of the country's school-children and youth population.”

Commentary #7 (Methods): Page 5, lines 119 - 120: Could the authors please state why the expert board and teachers felt that students under the age of 10 would be unable to complete the questionnaire accurately? Is there a reason that children aged 14 were not invited to participate in the study? My understanding is that the Canadian version was intended for students in grades 4 – 8, who could be between the ages of 8 – 14 years.

Page 6, lines 127 – 133:

We added three sentences: “Based on the recommendations of both the expert board and their teachers, the lower age limit was moved from eight to ten years—in consideration of the children's cognitive maturity, the text's content, the level of reading comprehension, and the ability to concentrate in children of this age. The maximum age limit was set to the age of 13. We estimated that if the questionnaire was comprehensible to this age group, it could be assumed that older pupils would find it understandable as well, based on their cognitive maturity and the subsequent increase in the level of the aspects mentioned above.”

Commentary #8 (Results): Page 7, lines 177 – 179: Are there students that participated in both phases of the study as 169 + 63 takes me above the 223 children stated in line 177. Were the 14 excluded subjects from both phase 1 and 2? If the students that did the accelerometry in phase 1 are different from those who did re-test reliability in phase 2, I think this must be made clear in the methodology and the reasoning explained.

1. Page 8, lines 190 – 191: We corrected the number of children included in the study.

2. Page 8, line 193: We added the sentence “26 children participated in both phases.”

3. Page 8, line 194: We corrected number of excluded children (“52”).

Commentary #9 (Results): Page 11, line 221: not statistically significant needs to replace statistically non-significant. They imply different things. We cannot claim that it is non-significant, only that it isn’t significant.

Page 11, lines 236 – 237: 

We replaced the sentence “statistically non-significant” with “not statistically significant”. 

Commentary #10 (Results): Table 1: Phase 2, n does not include the number of girls.

Table 1: 

We added the number of participated girls. 

Commentary #11 (Results): Table 2: Phase 2 includes concurrent validity (the 139 students that did both the accelerometry data and PAQ-C). According to the methods section on page 6, this is part of phase 1.

Table 2: 

We corrected this in accordance with commentary.

Commentary #12 (Discussion): Page 11, line 232: How does this questionnaire differ from the other 2 stated questionnaires? Is it shorter? Does it require less time? Is it more or less difficult? Please state the direction as readers may not be familiar with the other two questionnaires.

Page 11 and 12, lines 250 – 256: 

We replaced the original sentence “The outputs, length, time requirements, and overall difficulty of the PAQ-C/CZ differ from the other two questionnaires adapted into the Czech language (Health Behaviour in School-aged Children Questionnaire and Youth Activity Profile questionnaires) [7,8,10]” with four sentences “The PAQ-C/CZ varies from the other two questionnaires adapted into the Czech language (Health Behaviour in School-aged Children Questionnaire and Youth Activity Profile questionnaires) in several aspects [10–13]. First, The PAQ-C/CZ is considerably shorter, less difficult, requires less time, and therefore the children can concentrate when fulfilling the questionnaire. Second, the questionnaires’ outputs also differ. Unlike the PAQ-C/CZ, the first mentioned questionnaire is focused on a comprehensive assessment of healthy behavior and the second one on the estimation of values comparable to the objective PA measurement.”

Commentary #13 (Discussion): Page 12, lines 251 – 253: Do the authors have any thoughts as to why there was no difference between boys and girls in this population? This was also shown in your accelerometry data for MVPA and could add to the discussion.

Page 12, lines 274 – 278: 

We added four new sentences to the discussion: “In the Czech Republic, there has been no evidence of objectively evaluated PA based on systematic research in this age group so far. Therefore, it is virtually impossible to provide further interpretation of the findings. For instance, it is not possible to determine whether they are specific for Czech children of this age in general or whether the actual recruitment of participants caused a distortion. That said, the findings present a welcome challenge to conduct further research.”

Commentary #14 (Discussion): Page 15, line 320: I would caution the authors on using the words “identical methods” as it implies all the details are the same, which is not true. For example, the Polish version and the Chinese version used 5s epochs to quantify MVPA by accelerometry, which differs to the 15s used here. I would recommend “followed similar protocols”.

1. Page 15, lines 337 – 338: 

We replaced part of the sentence “…of the PAQ-C employing the same methods…” with “…of the PAQ-C followed similar protocols…”.

2. Page 16, lines 369 – 370: 

We replaced part of the sentence “…studies validating PAQ-C; this is due to the use of identical methods.” with “…studies validating PAQ-C which followed similar protocols.”

Commentary #15 (Discussion): Page 15, lines 326 – 332: Here the authors explain why children below 10 were not used in the test, but is there a reason that 14-year-old children were also excluded? Could the authors please explain.

Page 16, lines 382 – 384: 

We replaced a part of the original sentence “Taking that into consideration, it was consequently decided that the PAQ-C/CZ will only be used with children older than 10 years.” with “Taking that into consideration, it was consequently decided that the PAQ-C/CZ will only be used with children 10+ years old.“ 

Furthermore, we added two new sentences “Moreover, 14-year-olds were not included in the study. When selecting the sample, we estimated that if the questionnaire was comprehensible to the children aged 10–13 years, older pupils would find it understandable as well.”

Commentary #16 (Conclusion): Page 15, line 341: The first 4 words of the sentence do not flow into the rest of the sentence.

Page 17, lines 391 – 392: 

We reworded the first part of the original sentence “The most common questionnaire psychometric properties were evaluated including internal consistency, item/scale relationship, factor structure, test-retest reliability, and concurrent validity.” with “The psychometric properties of the questionnaire were evaluated, including internal consistency, item/scale relationship, factor structure, test-retest reliability, and concurrent validity.”

RESPONSES TO REVIEWER #2:

Commentary #1: The results of the PAQ-C/CZ indicated satisfactory internal consistency (Cronbach’s alpha = 0.77), not good.

Page 13, line 278: 

We modified the sentence “The results of Cronbach’s alpha (0.77) and CITCs (range between 0.29–0.61) showed a good internal consistency and acceptable item/scale properties of the PAQ-C/CZ [37].“ in this way “The results of Cronbach’s alpha (0.77) and CITCs (range between 0.29–0.61) indicated satisfactory internal consistency and acceptable item/scale properties of the PAQ-C/CZ [37] .”

Commentary #2: The item/scale properties indicated diverse item-total correlations (corrected item-total correlations ranged between 0.29–0.61). It needs more explanations and justifications.

Page 13, lines 279–281: 

We added two sentences “However, the wide range of observed CITCs shows the scale is not fully homogeneous. Especially, the values in the first four items indicate they contribute less to the total PAQ-C/CZ score.” and we have changed the beginning of the next sentence from “The observed levels….” to “Nevertheless, the observed levels…”.

Commentary #3: The factor loading of the exploratory factor analysis ranging from 0.30 to 0.79 which can't demonstrate the acceptance level in factor analysis for this study.

Page 13, lines 286–290 and 296–302: 

We corrected and extended the discussion related to exploratory factor analysis by “The observed model showed factor loads ranging widely from 0.32 to 0.73, whereas items before-school activity, physical education, and recesses only contribute to the total score a little. This probably corresponds to low variability explained by the factor (37.0%) and suggests there are still many other latent variables explaining the total score in the questionnaire. Factor load in four items is under 0.50, which is low.” and “In accordance with that structure, the lowest factor load was also found in the item before school and those items related to school (recesses and Physical Education); this could indicate the presence of another factor related to these items. Although the tendency of the items’ factor load is close to those presented in the mentioned studies, factor loads in the first four items are under 0.50, which is low. Unfortunately, the data did not provide evidence of another factor that would better describe the structure of the questionnaire. It would be desirable for further research to suggest how to strengthen the factor structure.” This correction and additional information should prevent misinterpretation of the results.

Additional explanation: 

It was found that extending the analysis by another factor (its eigenvalue = 0.997) will increase explained variability up to 48.1%. However, the subsequent analysis did not prove to be beneficial in terms of the identified factor loads. 

We believe that this information will be helpful to clarify our response. However, we do not think it is appropriate to include it in the results section.

Commentary #4: The concurrent validity between the PAQ-C/CZ total score and the accelerometer-determined MVPA is 0.29 which is quite low and can't reflect the objective PA level by the self-reported questionnaire through this study.

1. Page 14, line 328: 

We changed interpretation of 0.29 (0.28 after corrections mentioned above). In the sentence “The value is a little lower in comparison to the study reporting concurrent validity of the original PAQ-C version (rho = 0.39) in which Caltrac accelerometer was used as a concurrent method” we replaced “a little lower” by “…lower…”.

2. Page 17, lines 397 – 401: 

In accordance with all previous corrections and changes, we replaced two sentences in this section. 

“Concurrent validity analysis indicated that the PAQ-C/CZ could assess MVPA with acceptable validity. This instrument could also be used in future large-sample studies. However, given the limitations mentioned above, further research is necessary.” was replaced by “Borderline concurrent validity indicated that the PAQ-C/CZ can assess MVPA with certain limitations in children. Considering all the observed psychometric properties, it is recommended to use PAQ-C/CZ in large-sample research studies only and with emphasis on a correct interpretation of results. Given the limitations mentioned above, further research is required in order to improve the psychometric properties of the questionnaire, and to make further adjustments to it if necessary.”

---

## [Decision Letter · Decision Letter 1]

28 Dec 2020

Adaptation and validation of the Physical Activity Questionnaire for Older Children (PAQ-C) among Czech children

PONE-D-20-26808R1

Dear Dr. Cuberek,

We’re pleased to inform you that your manuscript has been judged scientifically suitable for publication and will be formally accepted for publication once it meets all outstanding technical requirements.

Kind regards,

Peter M ten Klooster, Ph.D.

Academic Editor

PLOS ONE

Additional Editor Comments (optional):

The revision has been reviewed by one of the original reviewers, who indicated that all comments were adequately addressed.

I also reviewed the manuscript, specifically focussing on the issues raised by Reviewer 2, which are adequately incorparated in the both the response and revised manuscript. The manuscript now provides a more critical and realistic interpretation of the actual psychometric findings.

Reviewers' comments:

Reviewer's Responses to Questions

**Comments to the Author**

1. If the authors have adequately addressed your comments raised in a previous round of review and you feel that this manuscript is now acceptable for publication, you may indicate that here to bypass the “Comments to the Author” section, enter your conflict of interest statement in the “Confidential to Editor” section, and submit your "Accept" recommendation.

Reviewer #1: All comments have been addressed

2. Is the manuscript technically sound, and do the data support the conclusions?

Reviewer #1: Yes

3. Has the statistical analysis been performed appropriately and rigorously? 

Reviewer #1: Yes

4. Have the authors made all data underlying the findings in their manuscript fully available?

Reviewer #1: Yes

5. Is the manuscript presented in an intelligible fashion and written in standard English?

Reviewer #1: Yes

6. Review Comments to the Author

Reviewer #1: (No Response)

7. PLOS authors have the option to publish the peer review history of their article (what does this mean?). If published, this will include your full peer review and any attached files.

Reviewer #1: No

---

## [Editor Report · Acceptance letter]

4 Jan 2021

PONE-D-20-26808R1 

Adaptation and validation of the Physical Activity Questionnaire for Older Children (PAQ-C) among Czech children 

Dear Dr. Cuberek:

I'm pleased to inform you that your manuscript has been deemed suitable for publication in PLOS ONE. Congratulations! Your manuscript is now with our production department. 

Kind regards, 

on behalf of

Dr. Peter M ten Klooster 

Academic Editor

PLOS ONE